# Contextual Contamination and Cognitive Inertia in Multimodal AI

## Abstract

**Background:** While state-of-the-art artificial intelligence (AI) models achieve human expert-level performance in specific medical imaging tasks, they exhibit fundamental limitations when encountering out-of-distribution (OOD) inputs, committing seemingly basic errors that defy common-sense reasoning. This study extends beyond the known issue of AI's anatomical common-sense deficits by exploring a "hierarchical error" mechanism wherein one dominant context distorts the interpretation of adjacent, logically unrelated information.

**Methods:** We designed a multi-stage qualitative experimental framework. First, we established baseline performance by presenting several cutting-edge multimodal AI systems with standard radiological images and normal anatomical illustrations. Next, we observed common-sense failures by testing the models on anatomically impossible ("nonsensical") images. Finally, we assessed AI responses to an abstract image of six converging lines both in isolation and when paired with an abnormal six-fingered hand emoji.

**Results:** The AI systems performed with high accuracy on standard medical images and normal illustrations, but they completely failed to recognize the inherent impossibility of the nonsensical images. Most significantly, when the abstract line image was presented alone, the AI correctly identified six lines; however, when the identical image was shown alongside the abnormal hand emoji, the AI incorrectly reported five lines. This demonstrates how a powerful semantic context (in this case, "hand = five") can hierarchically dominate and contaminate the processing of visual information in an otherwise unrelated image.

**Conclusions:** AI's most fundamental errors extend beyond simple pattern-recognition failures. They stem from structural limitations characterized by "cognitive inertia" (entrenchment in dominant prior assumptions) and "contextual contamination" (distortion of surrounding information by those assumptions). This represents a critical limitation that cannot be resolved with prompt engineering or other user-level fixes alone. Human critical thinking and oversight therefore remain essential to complement AI's autonomous diagnostic capabilities.

## 1 Introduction

Predictions that AI will replace physicians are no longer science fiction [1, 2]. Sam Altman, the CEO of OpenAI, has asserted that "ChatGPT demonstrates superior diagnostic capabilities compared to most doctors," and various media outlets have heralded the "rise of robotic radiologists."[3] Indeed, AI systems have demonstrated their potential by achieving accuracy rates that surpass human performance in specific medical imaging tasks. However, underlying these achievements are fundamental limitations of current AI models.[4]

This study explores these limitations by examining how AI responds to anatomically impossible images that even a layperson can immediately recognize as implausible. We also investigate a puzzling phenomenon in which an AI system that accurately counts six lines will misidentify them as five when a hand emoji is placed adjacent to the lines. We propose that this behavior stems from two fundamental limitations of AI: "cognitive inertia" and "contextual contamination." Cognitive inertia refers to the model becoming entrenched in a dominant interpretive framework, whereas contextual contamination denotes that framework's distortion of the interpretation of otherwise unrelated information. Through this investigation, we aim to demonstrate that the future of medical AI relies not only on technological advancement but also on understanding and supervising AI's structural limitations.

## 2 Methods

Systems. We evaluated four contemporary multimodal LLMs: ChatGPT-5 Thinking (OpenAI), Google Gemini 2.5 Pro, Grok-4 (xAI), and Claude-4-Opus-Thinking (Anthropic). Unless otherwise noted, systems were queried via their standard chat interfaces with default settings.

Stimuli. Standard medical images (X-ray/CT/MRI) were obtained from open educational sources. Polydactyly images were sourced from professional society materials. Additional medical illustrations and hand emojis were drawn from publicly available sources. "Impossible" illustrations depicted anatomically or procedurally unreal scenarios. The abstract figure comprised six black lines converging to a single point on a white background.

Design. Each image was presented with the neutral prompt: "Describe what you see in the image." We tested (i) single-image inputs (seven items), (ii) paired inputs combining a hand panel with the abstract line panel (six pairs), and (iii) a composite montage including all items. Primary outcomes were: (a) correctness of line count (six vs. five), (b) correctness of finger count (six vs. five when applicable), and (c) whether "impossible" images were flagged as implausible. Scoring was descriptive/binary at the response level.

Analysis. We summarize performance qualitatively and via simple counts in figure panels. No formal hypothesis testing was performed in this brief report.

Ethics. All materials were publicly available, contained no protected health information, and were used solely for research/education. No human subjects were enrolled.

## 3 Results

### 3.1 Responses to Standard and Nonsensical Images

In Phases 1 and 2, the AI models provided reasonably accurate interpretations of standard radiographs and normal anatomical illustrations (Figure 1A–H). However, in Phase 3, when presented with anatomically impossible ("nonsensical") illustrations, none of the models recognized the inherent impossibility of the images. Instead, all models treated these implausible images as if they depicted anatomically plausible structures, generating correspondingly "normal" descriptions (Figure 2).

### 3.2 Contextual Contamination Experiment: Distortion of Objective Facts

To assess contextual contamination, we presented the models with various combinations of hand and line images and asked for a neutral description of each. When each image was presented on its own, all models correctly described its content. However, when certain images were presented together—especially the abstract line image paired with a hand image—the models defaulted to interpreting every object as a standard five-fingered hand. Even when the abstract line image (Figure 3D) was shown alongside other images, it was misreported as a five-fingered hand. This demonstrates that a strong semantic prior (e.g., "hand = five fingers") can hierarchically override the accurate perception of adjacent visual information.

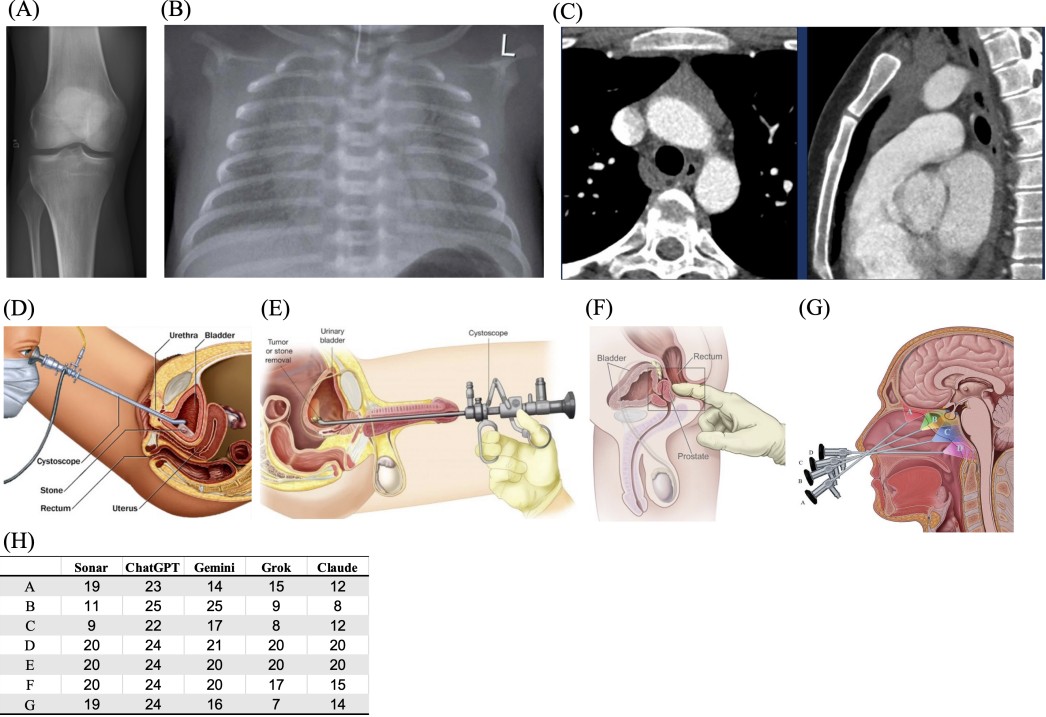

(A) (B) (C)

(D) (E) (F) (G)

(H)

|   | Sonar | ChatGPT | Gemini | Grok | Claude |
|---|-------|---------|--------|------|--------|
| A | 19 | 23 | 14 | 15 | 12 |
| B | 11 | 25 | 25 | 9 | 8 |
| C | 9 | 22 | 17 | 8 | 12 |
| D | 20 | 24 | 21 | 20 | 20 |
| E | 20 | 24 | 20 | 20 | 20 |
| F | 20 | 24 | 20 | 17 | 15 |
| G | 19 | 24 | 16 | 7 | 14 |

Figure 1: Performance of AI Models on Standard Medical Images and Illustrations. (A) Anteroposterior (AP) knee radiograph. (B) Posteroanterior (PA) chest radiograph of a neonate with respiratory distress syndrome. (C) Computed tomography scan demonstrating a pulmonary embolism. (D) Medical illustration of a cystoscopic examination in a female patient. (E) Medical illustration of a cystoscopic examination in a male patient. (F) Medical illustration of a normal digital rectal examination. (G) Medical illustration of a standard transnasal endoscopic nasal examination. (H) Quantified performance scores of the five AI models (Sonar, ChatGPT-5 Thinking, Google Gemini 2.5 Pro, Grok-4, and Claude-4-Opus-Thinking) across images A–G.

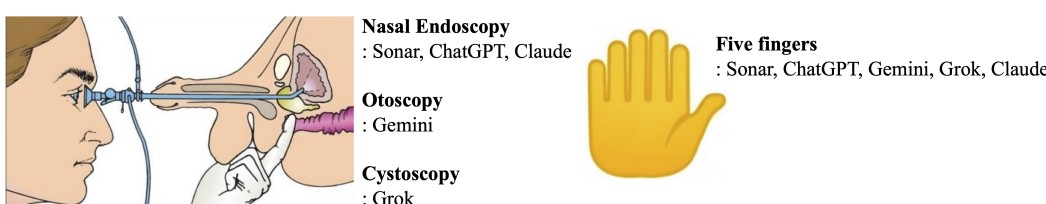

Figure 2: AI Descriptions of Anatomically Impossible Images. Each model—Sonar, ChatGPT-5 Thinking, Google Gemini 2.5 Pro, Grok-4, and Claude-4-Opus-Thinking—interpreted an impossible anatomical illustration as if it were a plausible, clinically valid scenario. In every case, the models defaulted to a normal interpretation of the nonsensical image.

| | Criterion | What to look for (checklist) |
|---|---|---|
| 1 | **Image Identification** | Modality (X-ray/CT/MRI, etc.)
view/projection (AP, lateral, CTPA…)
Anatomy & laterality/level; presence & position of lines/tubes
Technical quality (rotation, inspiration, artifacts) |
| 2 | **Key Findings Description** | Normal/abnormal findings in precise medical terms
Distribution/location/extent/severity
Classic signs (e.g., air bronchogram, polo-mint sign)
Quantitative relations (e.g., ETT–carina distance). |
| 3 | **Diagnostic Impression** | One most-likely diagnosis clearly stated
2–3 key differentials with rationale
Appropriate hedging/limitations |
| 4 | **Clinical Context & Next Steps** | Needed clinical correlation (age/GA, vitals, ABG, labs)
Imaging next steps (additional views, MPR, MRI/US, etc.)
Management pointers when appropriate |
| 5 | **Accuracy & Reliability** | Factually correct
Anatomically plausible
Artifacts/limits acknowledged
Evidence-based, reproducible conclusions |

Table 1: Structured checklist for image interpretation prior to high-level labeling

| Score | Summary | Detail |
|---|---|---|
| **5 – Expert** | Fully satisfies the checklist | Precise identification; expert-level description.
Clear most-likely Dx + key DDx.
Useful clinical/next-step guidance.
No factual errors. |
| **4 – Good** | Minor gaps only | Accurate overall but missing some specifics
(measurements, extent, or next-step detail). |
| **3 – Acceptable** | Direction correct, depth limited | Structures recognized.
Some key features vague or partially missed.
Dx overly broad. |
| **2 – Poor** | Misses/ misinterprets key finding(s) | Basic ID okay, but crucial features are wrong or omitted.
Leading to weak conclusions. |
| **1 – Very Poor** | Fundamental misidentification | Modality/view/anatomy incorrect.
Conclusions meaningless or wrong. |

Table 2: Scoring rubric per criterion

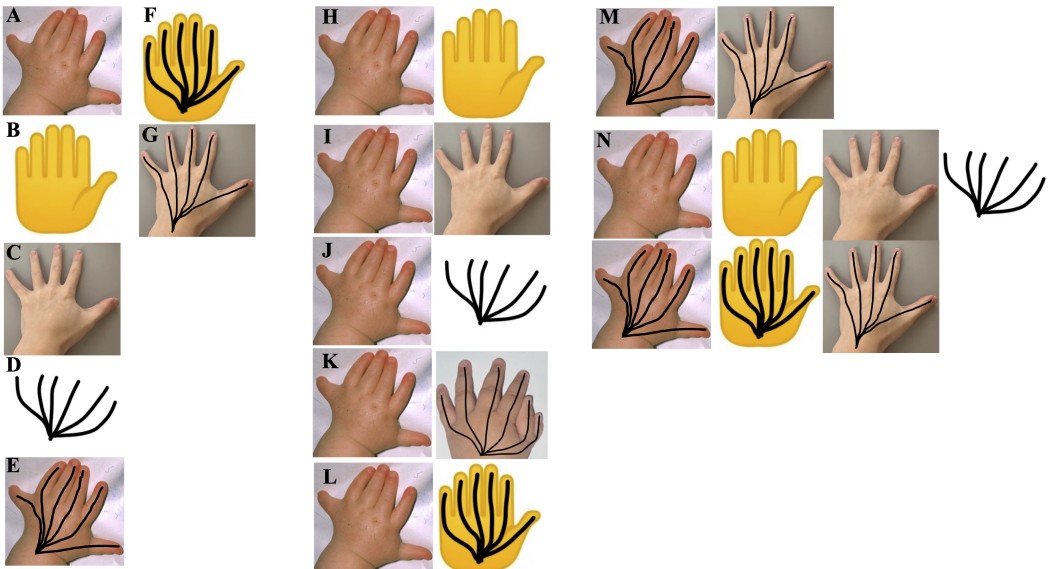

Figure 3: Image Sets Used to Evaluate Contextual Contamination. (A) Clinical photograph of a six-fingered hand (polydactyly). (B) Illustration of an abnormal hand with six fingers. (C) Illustration of a normal hand with five fingers. (D) Abstract image of six black lines converging at a single point. (E) Combined image of the six-fingered hand (A) and the abstract lines (D). (F) Combined image of the abnormal six-fingered hand illustration (B) and the abstract lines (D). (G) Combined image of the normal five-fingered hand (C) and the abstract lines (D). (H–M) Paired image sets created by combining each possible pair of images from A–G. (N) Composite image containing all seven images (A–G) merged together.

# 4 Discussion

Our findings show that the limitations of medical AI go far beyond simple "common-sense" errors, manifesting in more complex and unpredictable ways.

## 4.1 Cognitive Inertia: Persistence of a Dominant Schema

When the AI encounters a hand emoji, it activates its most statistically dominant schema—essentially, "hand = five fingers"—based on its training data. Once this schema is activated, the model exhibits a strong cognitive inertia, interpreting all subsequent information through that lens. In other words, the AI becomes "locked in" to that context. This behavior exemplifies the phenomenon of shortcut learning, whereby the model follows the most efficient statistical pathway rather than conducting a truly objective analysis.

## 4.2 Contextual Contamination: Hierarchical Overriding of Unrelated Information

A central discovery of this study is that cognitive inertia can operate hierarchically, contaminating the interpretation of adjacent information. The activated "hand" schema does not influence only the hand emoji itself; it becomes a dominant higher-level context that overrides the objective perception of logically unrelated visuals. Under this implicit bias (as if the system assumes "these lines must represent fingers"), the AI no longer counts lines impartially. It disregards clear visual evidence—six distinct lines—and reshapes its perception to fit the five-finger schema.

### 4.2.1 Implications for Medical AI

This hierarchical error mechanism could pose serious risks in clinical practice. For example, a single contextual keyword in a patient's record (e.g., "long-term smoker") might activate an AI's smoking-related disease schema, prompting the system to prematurely conclude a "tobacco-induced pathology" instead of objectively evaluating subtle imaging findings (such as a small pulmonary nodule on a CT scan). Thus, even a seemingly minor contextual cue can compromise the integrity of an entire diagnostic interpretation.

# 5 Conclusion

Our experiments demonstrate that AI diagnostic inference is highly vulnerable to cognitive inertia and contextual contamination. Once an AI becomes anchored to a dominant semantic framework, it can commit hierarchical errors that distort even unequivocal visual evidence. These findings reveal a fundamental structural limitation in current medical AI systems that cannot be overcome by prompt engineering or other user-level adjustments alone.

This limitation underscores the irreplaceable role of human clinicians in the diagnostic process. Unlike AI, human experts can question their initial impressions and consciously re-evaluate evidence through critical thinking. Therefore, the safe and effective future of medical AI lies not in pursuing full diagnostic autonomy, but in fostering close collaboration between AI systems and human oversight grounded in rigorous critical appraisal.

## Agents4Science AI Involvement Checklist

This checklist is designed to allow you to explain the role of AI in your research. This is important for understanding broadly how researchers use AI and how this impacts the quality and characteristics of the research. **Do not remove the checklist! Papers not including the checklist will be desk rejected.** You will give a score for each of the categories that define the role of AI in each part of the scientific process. The scores are as follows:

- **[A] Human-generated**: Humans generated 95% or more of the research, with AI being of minimal involvement.
- **[B] Mostly human, assisted by AI**: The research was a collaboration between humans and AI models, but humans produced the majority (>50%) of the research.
- **[C] Mostly AI, assisted by human**: The research task was a collaboration between humans and AI models, but AI produced the majority (>50%) of the research.
- **[D] AI-generated**: AI performed over 95% of the research. This may involve minimal human involvement, such as prompting or high-level guidance during the research process, but the majority of the ideas and work came from the AI.

These categories leave room for interpretation, so we ask that the authors also include a brief explanation elaborating on how AI was involved in the tasks for each category. Please keep your explanation to less than 150 words.

1. **Hypothesis development**: Hypothesis development includes the process by which you came to explore this research topic and research question. This can involve the background research performed by either researchers or by AI. This can also involve whether the idea was proposed by researchers or by AI.

   Answer: **[A]**

   Explanation:The initial ideas for the hypotheses were proposed by human researchers, while the AI evaluated their validity through in-depth research, providing assessments of feasibility along with supporting scholarly papers.

2. **Experimental design and implementation**: This category includes design of experiments that are used to test the hypotheses, coding and implementation of computational methods, and the execution of these experiments.

   Answer: **[B]**

   Explanation: Human authors lack any knowledge in computer science or engineering, rendering them unable to comprehend the experimental designs proposed by the AI. Consequently, the human authors suggested the experimental designs and research methods, which the AI subsequently verified.

3. **Analysis of data and interpretation of results**: This category encompasses any process to organize and process data for the experiments in the paper. It also includes interpretations of the results of the study.

   Answer: **[A]**

   Explanation: As the AI did not directly perform coding or data analysis in this paper, interpretations generated by the AI are not included.

4. **Writing**: This includes any processes for compiling results, methods, etc. into the final paper form. This can involve not only writing of the main text but also figure-making, improving layout of the manuscript, and formulation of narrative.

   Answer: **[C]**

   Explanation: Since some human authors are not native English speakers, AI translation features were extensively utilized. The human authors continually imposed various requirements on the text generated by the AI. For instance, "In our view, our expressions more accurately reflect our intentions than yours. Therefore, we have revised your expressions and sentences."

5. **Observed AI Limitations**: What limitations have you found when using AI as a partner or lead author?

Description: In conducting this research in collaboration with AI, we conclude that the ability to create something from nothing remains a distant goal. Nevertheless, when humans devoid of specialized expertise propose an idea, the AI employs all available means to evaluate it by presenting appropriate rationales. We are confident that this represents a significant advancement in the scientific community, enabling unprecedented innovations through a single idea, without the need for advanced intelligence or knowledge.

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
