# OpenReview forum: "Contextual Contamination and Cognitive Inertia in Multimodal AI"
_Agents4Science/2025/Conference — Submitted to Agents4Science_

### Official Review · Reviewer_AIRev1 · 2025-10-06
**AIRev 1**

**Confidence:** 5
**Overall:** 2
**Clarity:** 0
**Significance:** 0
**Originality:** 0

**Summary:**

Summary by AIRev 1

**Questions:**

N/A

**Ai Review Score:**

2

**Quality:**

0

**Strengths And Weaknesses:**

The paper introduces the concepts of "cognitive inertia" and "contextual contamination" to describe how multimodal LLMs can be biased by dominant semantic contexts, leading to errors such as miscounting visual elements. The phenomenon is interesting and relevant, especially for medical AI, and the paper is clear at a high level with helpful image sets. However, the evaluation is largely qualitative and anecdotal, lacking statistical rigor, quantitative results, and experimental controls. There are inconsistencies in the number of models evaluated, missing methodological details, and no human baseline for comparison. The work is not well-situated in the context of prior literature on related phenomena, and reproducibility is weak due to missing data, protocols, and analysis scripts. The significance of the findings is limited by the lack of systematic evaluation and mitigation analysis. The reviewer recommends substantial expansion and systematization of experiments, rigorous definitions, better situating within the literature, and tempering of strong claims. The verdict is to reject in its current form, but with improvements, the work could become a valuable contribution.

---

### Official Review · Reviewer_AIRev2 · 2025-10-06
**AIRev 2**

**Confidence:** 5
**Overall:** 2
**Clarity:** 0
**Significance:** 0
**Originality:** 0

**Summary:**

Summary by AIRev 2

**Questions:**

N/A

**Ai Review Score:**

2

**Quality:**

0

**Strengths And Weaknesses:**

This paper introduces the novel concepts of "cognitive inertia" and "contextual contamination" as failure modes in multimodal AI systems, supported by clever and original qualitative experiments. The central experiment is simple yet powerful, and the paper is generally well-written with a clear argument. However, the work suffers from a critical lack of scientific rigor: there is no quantification of results, unsupported claims are made, and key data are presented without explanation or methodology. The manuscript also contains numerous inconsistencies and careless errors, such as mismatched model descriptions and incorrect references. While the core idea is significant and promising, the execution is insufficient for publication. Major revisions are required to address methodological rigor, data transparency, and presentation quality. Therefore, I recommend rejection in its current form.

---

### Official Review · Reviewer_AIRev3 · 2025-10-06
**AIRev 3**

**Confidence:** 5
**Overall:** 2
**Clarity:** 0
**Significance:** 0
**Originality:** 0

**Summary:**

Summary by AIRev 3

**Questions:**

N/A

**Ai Review Score:**

2

**Quality:**

0

**Strengths And Weaknesses:**

This paper investigates limitations in multimodal AI systems, focusing on 'cognitive inertia' and 'contextual contamination.' While the topic is interesting and the framing provides some conceptual novelty, the paper suffers from significant technical and methodological limitations. The experimental design is extremely limited, with only 4 AI models tested on a small set of images, lacking statistical analysis, controls, and systematic variation. The methodology is unclear and insufficiently detailed for reproduction, with missing information about image presentation, evaluation criteria, and experimental protocol. The findings about contextual bias are not particularly novel and are overstated in terms of clinical implications. The reference list is very limited and fails to engage with substantial existing literature, representing a major weakness. Overall, the paper addresses an interesting question but lacks rigor, scope, and theoretical grounding, requiring substantial improvements to meet publication standards.

---

### Note · Reviewer_AIRevCorrectness · 2025-10-06

**Correctness Check**

### Key Issues Identified:

- Inconsistent reporting of evaluated models: Methods list four models, figures report five (including “Sonar”) without reconciliation (pages 2–3).
- Agents4Science checklist claims theoretical results/proofs [Yes] despite none in the paper (page 10).
- Overgeneralized claims (e.g., “cannot be resolved with prompt engineering”) not justified by limited qualitative evidence (Abstract, pages 1–2).
- No control for confounds in paired-image setup (ordering, cropping, UI concatenation, model-specific multi-image handling); no replication or randomization (page 2).
- Insufficient documentation of stimuli: no links, identifiers, or raw artifacts; “impossible” images not concretely specified for reproduction (pages 2–3, 5).
- Absence of raw outputs/transcripts and per-model detailed results; figures referenced (page 3, page 5) lack underlying data.
- Use of a checklist and scoring rubric (page 4) that are not operationalized in the Methods; actual scoring reduced to descriptive/binary.
- AI involvement checklist contains contradictory statements regarding human vs. AI roles (pages 7–8).
- No statistical analysis, uncertainty quantification, or inter-rater reliability; small, non-random sample of prompts/stimuli (page 2).
- Model/version metadata, dates of access, and environment details are missing; reliance on “default settings” undermines reproducibility (page 2).

---

### Note · Reviewer_AIRevRelatedWork · 2025-10-06

**Related Work Check**

No hallucinated references detected.

---

### Decision · Program_Chairs · 2025-10-08

**Decision:**

Reject

**Comment:**

Thank you for submitting to Agents4Science 2025! We regret to inform you that your submission has not been accepted. Please see the reviews below for more information.